# Dysregulated Sulfide Metabolism in Multiple Sclerosis: Serum and Vascular Endothelial Inflammatory Responses

**Pooja Veerareddy [1,†], Nhi Dao [1,†], Jungmi W. Yun [1], Karen Y. Stokes [1,2], Elizabeth Disbrow [2,3], Christopher G. Kevil [1,4], Urska Cvek [5], Marjan Trutschl [5], Philip Kilgore [5], Murali Ramanathan [6], Robert Zivadinov [6,7] and Jonathan S. Alexander [1,2,3,\*]**

[1] Department of Molecular and Cellular Physiology, LSU Health Shreveport, Shreveport, LA 71103, USA
[2] Center for Brain Health, LSU Health Shreveport, Shreveport, LA 71130, USA
[3] Department of Neurology, LSU Health Shreveport, Shreveport, LA 71130, USA
[4] Department of Pathology, LSU Health Shreveport, Shreveport, LA 71130, USA
[5] Department of Computer Science, LSU-Shreveport, Shreveport, LA 71115, USA
[6] Department of Pharmaceutical Sciences, Jacobs School of Medicine and Biomedical Sciences, University at Buffalo, Buffalo, NY 14203, USA
[7] Department of Neurology, Buffalo Neuroimaging and Sciences (BNAC), Jacobs School of Medicine and Biomedical Sciences, University at Buffalo, Buffalo, NY 14203, USA
\* Correspondence: jalexa@lsuhsc.edu; Tel.: +1-318-675-4151
† These authors contributed equally to this work.

**Abstract:** Multiple sclerosis (MS) is a leading cause of neurodegenerative disability in younger individuals. When diagnosed early, MS can be managed more effectively, stabilizing clinical symptoms and delaying disease progression. The identification of specific serum biomarkers for early-stage MS could facilitate more successful treatment of this condition. Because MS is an inflammatory disease, we assessed changes in enzymes of the endothelial hydrogen sulfide (H$_2$S) pathway in response to inflammatory cytokines. Blotting analysis was conducted to detect Cystathionine $\gamma$-lyase (CSE), Cystathionine beta synthase (CBS), and 3-mercaptopyruvate sulfurtransferase (MST) in human brain microvascular endothelial apical and basolateral microparticles (MPs) and cells following exposure to tumor necrosis factor-$\alpha$ (TNF-$\alpha$) and interferon-$\gamma$ (IFN-$\gamma$). CSE was increased in MPs and cells by exposure to TNF-$\alpha$/IFN-$\gamma$; CBS was elevated in apical MPs but not in cells or basolateral MPs; MST was not significantly affected by cytokine exposure. To test how our findings relate to MS patients, we evaluated levels of CSE, CBS, and MST in serum samples from healthy control and MS patients. We found significantly decreased levels of CBS and MST ($p$ = 0.0004, 0.009) in MS serum samples, whereas serum levels of CSE were marginally increased ($p$ = 0.06). These observations support increased CSE and lower CBS and MST expression being associated with the vascular inflammation in MS. These changes in endothelial-derived sulfide enzymes at sites of inflammation in the brain may help to explain sulfide-dependent changes in vascular dysfunction/neuroinflammation underlying MS. These findings further support the use of serum samples to assess enzymatic biomarkers derived from circulating MPs. For example, "liquid biopsy" can be an important tool for allowing early diagnosis of MS, prior to the advanced progression of neurodegeneration associated with this disease.

**Keywords:** biomarkers; multiple sclerosis; microparticles; brain endothelial cells; MS serum samples

## 1. Introduction

Multiple sclerosis (MS) is a neurodegenerative disease associated with inflammation in the central nervous system [1]. A total of 2.3 million people worldwide suffer from this disease, and environmental and genetic factors contribute to the development of MS [2]. Magnetic resonance imaging (MRI) can detect sites of neurological lesions in the brain, often correlated with the clinical symptoms seen in patients. In most cases of MS, initial clinical symptoms are intermittent, and then progress to a series of chronic events, typically diagnosed as relapsing–remitting MS (RRMS) [1]. In MS, the immune system destroys the insulating sheath (myelin) enveloping nerve fibers, which impairs the efficiency of nerves in communicating and transmitting messages, usually at a greater energy expenditure [3]. There are no standard symptoms exhibited across all patients with MS; rather, symptoms depend on the severity of damage to white matter, and the locations of the affected nerves. Most patients diagnosed with MS (85%) initially have RRMS and experience exacerbation of new symptoms followed by a period of disease remission that can be brief or can last for several years [3]. Symptoms of MS commonly include limitation of movement with numbness or weakness in one side of the body at a time, diminished coordination, and vision problems. Approximately 50% of patients with relapsing–remitting MS endure a steady progression of intensifying symptoms with each relapse [3]. Although most individuals experience periods of disease remission, others have no remission throughout the process of symptom development. Instead, over time, up to 15% of MS patients will suffer irreversible progression of disease, termed secondary progressive MS (SPMS).

A dysregulation of the blood–brain barrier (BBB) is one of the first indications of cerebrovascular abnormalities seen in people diagnosed with MS [4]. The BBB is an organized complex, including cerebral endothelial cells, supporting pericytes, astrocytes, and glia, all of which govern the exchange of solutes and immune cells between the blood and the brain. MS progression disrupts the function of the BBB, resulting in abnormal inflammatory responses of the central nervous system and leading to vascular injury in the brain [5]. Current therapeutic measures can treat flare-ups in MS, but once a patient's diagnosis progresses to SPMS, there are fewer treatments which can arrest progressive neurodegeneration. Standard methods of MS diagnosis use MRI and spinal fluid analysis; however, these do not always provide accurate assessments of MS prognosis and do not distinguish between RRMS and SPMS. Early identification of easily measurable biomarkers in MS patients might allow for earlier diagnosis of MS before irreversible neurodegenerative deterioration occurs.

The purpose of this study was to evaluate changes in the sulfide-generating enzymes (CSE, CBS, and MST) in brain endothelial cells and in apical and basolateral microparticles (AMPs, BMPs) derived from these cells when exposed to inflammatory cytokines, as a model of MS inflammation. Evidence of proinflammatory responses initiated by the circulation of AMPs and BMPs has prompted the investigation of the presence, and the measurement of the levels, of sulfide-producing enzymes in the vascular space in order to determine whether there may be a correlation between increased levels of hydrogen sulfide and forms of neurodegeneration. This study was prompted by our findings in another form of neurodegeneration, Alzheimer's disease, where levels of circulating sulfides were significantly elevated in comparison with controls. Sulfides can be generated by CSE, CBS, and MST. Therefore, we sought to further study this in MS by evaluating the extent to which the levels of these circulating sulfide-generating enzymes are changed in serum from MS patients. Our goal was to identify a novel MS biomarker for diagnosing and possibly staging disease progression in MS.

A key feature of MS is neuroinflammation [6]. At sites of inflammation, e.g., in response to cytokines, brain endothelial cells may increase their release of MPs [7]. Brain endothelial cells release AMPs into the vascular space and BMPs into the perivascular space. Endothelial AMPs are regarded as biomarkers for vascular inflammation in the

brain [7]. Furthermore, the circulation of endothelial adhesion molecules and other proteins in various MPs derived from the parent cells may play an active role in the progression of MS by enhancing the proinflammatory responses of this disease [6]. Evidence of MP accumulation in the brain parenchyma suggests BMPs may also contribute to the progression of neurovascular inflammatory diseases. In support of this concept, we found that both AMPs and BMPs derived from cytokine-stimulated cells depress the contractility of brain vascular smooth muscle cells [6]. In MS, evaluation of lymphatic biomarkers in serum samples of patients with RRMS and SPMS revealed that human brain endothelial cells release neurolymphatic biomarkers within MPs [8]. These studies provide evidence that endothelial cells may intensify inflammation via MPs and support the development of these particles as important biomarkers and mediators of MS disease activity.

Hydrogen sulfide plays important roles in the regulation and vascular homeostasis and pathophysiological processes in the cerebral vasculature [9,10]. Recent research shows that sulfides may be dysregulated in dementia and could represent a possible biomarker for diagnosing neurovascular disturbances [9]. Cystathionine $\beta$-synthase (CBS), cystathionine $\gamma$-lyase (CSE), and 3-mercaptopyruvate sulfur transferase (MST) are three enzymes known to produce $H_2S$ [11,12]. All three enzymes are expressed in the brain, with MST being primarily responsible for $H_2S$ generation in the central and peripheral nervous systems. CBS is also expressed in the brain parenchyma, while CSE is the primary enzyme found in the brain vasculature [10,13]. CSE and CBS are cytosolic enzymes, whereas MST is localized in the mitochondria. $H_2S$ has been shown to reduce expression of many proinflammatory cytokines, chemokines, and enzymes in endothelial cells [14]. In the central nervous system, $H_2S$ increases the activity of NMDA receptors, which are critical for memory retention, and alleviates oxidative stress by modulating reactive oxidative species (ROS), commonly found at sites of inflammation in the brain [15]. Contrary to these beneficial functions of $H_2S$ in the vascular systems, evidence shows neurological stress and vascular dysfunction can be provoked by sulfide metabolites of $H_2S$ [9]. It is plausible that, like another gasotransmitter, nitric oxide, $H_2S$, and its metabolites may be beneficial at low concentrations, but at high levels they may accumulate and intensify the progression of neurodegenerative diseases [16–19]. In addition, sulfides exist in different forms, which may contribute to their protective/deleterious properties. Importantly, our group has reported that endothelial CSE appears to be a prominent source of polysulfides in tissues. In CSE-deficient mice, we found a *decrease* in blood–brain barrier permeability, indicating that CSE, a primary $H_2S$-producing enzyme in the endothelium, may diminish barrier function [18]. Furthermore, we showed that elevated sulfide in the circulation serves as a biomarker for disease activity (cognitive impairment) and vascular disturbances (lesion volume) seen in Alzheimer's disease and related dementias (ADRD) [9]. Lesion volume is also used to assess MS severity because it relates to "bradyphrenia", which can slow thoughts and delay responses in neurodegenerative diseases and disorders [20].

Although there are treatments to suppress the severity of MS symptoms, no methods completely arrest the degenerative processes seen in MS, and early intervention and treatment of RRMS prior to progression into secondary progressive MS is a keystone of MS therapy. MRI imaging and spinal fluid analysis can be inaccurate (and delayed) predictors of MS progression. Therefore, we sought to evaluate whether sulfide-generating enzymes can be used as biomarkers for MS, possibly representing early inflammatory changes in the brain vasculature. Our measurement of these enzymes released into AMPs or BMPs provide further insight into how they may serve as a circulating biomarker of what is delivered into the brain parenchyma by inflamed endothelial cells and could help to explain how endothelial inflammation provokes perivascular inflammatory phenomena. Once validated, biomarkers originating from brain endothelial cells themselves could provide a method for identifying sulfide-producing enzymes as they relate to MS. Further investigations may help to specify which enzyme or panel of enzymes in combinations may be present that contribute to the central sulfide burden of the brain. Establishment of

a simple and standardized procedure for identifying these enzymes as biomarkers in relapsing–remitting and secondary progressive MS could reinforce imaging and clinical/behavioral studies.

## 2. Materials and Methods

### 2.1. Clinical Specimens

Serum samples were collected under an IRB ('MS-Omics project RQ00328', HSIRB 435893-2, 9/6/13). Patients diagnosed with RRMS and SPMS and healthy control (HC) sample patients were enrolled at the Department of Neurology in the University of Buffalo, Buffalo, NY [8] (Table 1). Inclusion criteria were ages between 18 and 80 years with a diagnosis of RRMS or SPMS. MS was diagnosed according to the McDonald criteria and an MRI examination was performed less than 30 days of the clinical examination with the standardized study protocol. Patients were also required to have scores within the range of 0–8.5 on the Expanded Disability Status Scale (EDSS). Exclusion criteria included individuals who experienced a relapse or exacerbation of symptoms, had steroid treatment within less than 30 days of the start of this study, and had preexisting conditions associated with non-MS brain pathology or pregnancy. Healthy control subjects were obtained through hospital recruitment and advertisements. Physical examinations were performed, and individuals screened for autoimmune diseases, environmental and vascular risks, and personal habits capable of confounding this investigation and future studies. Blood was collected from all subjects, and MS serum samples were derived from blood collected from patients diagnosed with RRMS and SPMS. Serum is the remaining fluid following the centrifugation of blood to remove clots and blood cells. Frozen MS serum samples were collected from −80 degrees Celsius.

**Table 1.** Data of healthy controls and patients with multiple sclerosis.

|  | HC ($n = 60$) | MS ($n = 176$) | RRMS ($n = 150$) | SPMS ($n = 26$) |
|---|---|---|---|---|
| Sex, female, n (%) | 31 (52) | 136 (77) | 114 (76) | 22 (85) |
| Age, years, mean (SD) | 44 (14) | 46 (9.8) | 45 (9.6) | 55 (7) |

HC—healthy control; MS—multiple sclerosis; RRMS—relapsing–remitting multiple sclerosis; SPMS—secondary progressive multiple sclerosis.

### 2.2. Cell Culture

A polarized human cerebral microvascular endothelial cells line (hCMEC/D3) was cultured in flasks coated with rat tail collagen type I (0.1 mg/mL). The cell line was provided by Dr. Pierre-Oliver Couraud, Inserm, Paris, France. Growth medium consisted of an EndoGRO™—MV complete media kit from MiliporeSigma in Burlington, MA, USA. Cells in this experiment were cultured at 37 °C in 5% $CO_2$, and passages between 28 and 34 were used for experiments.

### 2.3. Cytokine Treatment

To evaluate endothelial microparticles, hCMEC/D3 were placed in 6-well plates with 3 μm inserts in complete growth media. A measure of 2 mL of medium was added to the bottom (basolateral chamber) of each well, and 1.5 mL of medium was added to the top (apical chamber) of the insert where cells were placed. Following a period of 48 h, cells reached confluency and media in both compartments were replaced with media containing 1000 U/mL of interferon-gamma (IFN-$\gamma$,) and/or 20 ng/mL of tumor necrosis factor-alpha (TNF-$\alpha$) (T/I) or control medium. TNF-$\alpha$ and IFN-$\gamma$ are cytokines known to play a role in multiple sclerosis [21]. Media were separately removed from apical and basolateral compartments to isolate MP at 72 h following treatment. All experiments were normalized to the surface area of D3 cells used to produce MPs.

### 2.4. Isolation of Endothelial Microparticles from D3 Cells

We isolated endothelial "microparticles" which are released from endothelial cells based on a previous approach that used calibrating microparticle flow cytometric analysis using 0.5, 1, and 2 μm FluorsbriteTM Yellow Green Microspheres (Polysciences, Warrington, PA, USA) [6]. Logarithmic-scale side-scatter plots of sizing beads were used to determine appropriate gating for the samples and found that particles isolated in this method were in the range of 0.5–1 μm in size. Therefore, we refer to these as "microparticles" because apoptotic bodies are larger in size (up to 5 μm) and exosomes are much smaller (smaller than 150 nm). We have only evaluated these particles and cannot exclude possible contributions of these other particle types which may be present and active in vivo.

Following exposure of D3 cells to control medium or medium supplemented with IFN-γ/TNF-α, supernatants from the apical and basolateral chambers were centrifuged at 400× $g$ for 10 min at 4 °C. Resulting supernatants were transferred to fresh micro centrifuge tubes and were centrifuged again at 20,800× $g$ for 1 h at 4 °C to pellet AMPs or BMPs from apical and basolateral chambers, respectively. Supernatants were aspirated and MP pellets were washed twice by centrifugation using 4 °C PBS plus 1 mM phenylmethylsulfonyl fluoride (PMSF) at 20,800× $g$ for a period of 15 min at 4 °C. Supernatants were then discarded and MP pellets were stored at −80°C. Cells were lysed in 200 μL of Laemmli sample buffer plus 1 mM PMSF, sonicated, and stored at −80 °C until immunoblotted.

### 2.5. Blotting Analysis

Frozen MP, cell, and MS serum samples were thawed, and 0.5 μL of each of the samples was loaded onto nitrocellulose membranes and left to dry overnight. Ponceau S staining was used for protein loading standardization. A measure of 5% milk and TBST were used to block (2 h) and rinse the blots (3X) after staining. Membranes were immunoblotted for cystathionine γ-lyase (CSE), cystathionine β-synthase (CBS), and 3-mercaptopyruvate sulfur transferase (MST) and incubated in Clarity Western Peroxide Reagent and visualized using Clarity Western Luminol/Enhancer Reagent (Biorad, Hercules, CA, USA). All primary antibodies were used at 1:500 dilution and incubated overnight. Secondary antibodies were used at 1:2000 dilution and incubated for 1 h at room temperature before washing and ECL reactions.

ChemiDoc imaging system (Biorad, Hercules, CA, USA) was used to develop images of membranes. Densitometry was performed using ImageJ analysis software, v.153 (NIH, Bethesda, MD, USA). Data were normalized to total protein using Ponceau S Staining (Sigma biochemicals, St. Louis, MO, USA).

### 2.6. Antibodies

Rabbit anti-Human Cystathionine-gamma-lyase (CSE) Polyclonal Antibody (cat. No. MBS2014844, MyBioSource, San Diego, CA, USA), Cystathionine-beta-synthase (CBS) (N-Term) Antibody (cat. No. ABIN629598, Antibodies Online, Limerick, PA, USA), Recombinant Anti-MST3 Antibody [EP1468Y] (cat. No. Ab51137, Abcam, Cambridge, MA, USA), and Anti-Rabbit IgG (whole molecule)-Peroxidase antibody produced in goat (cat. No. 028M4755V, Sigma-Aldrich, St. Louis, MO, USA) were used in blotting analysis procedures.

### 2.7. Statistical Analysis

Statistical analyses were performed on all datasets using a basic Student's unpaired, two-sided *t*-test. This analysis compared control samples to the cytokine-treated microparticle and cell samples in Figures 1–3. In Figure 4, the t-test compared HC to RRMS and SPMS condensed data (MS serum samples). The distribution of the enzymes within healthy control and RRMS and SPMS samples was characterized using violin plots (Figure 4); these were generated based on the results of the corresponding statistical test using the vioplot package for the R statistical language [22]. A Kruskal–Wallis one-way analysis of

variance (ANOVA) performed on condensed (MS), and uncondensed (RRMS and SPMS) datasets allow for comparison of the respective datasets to the HC samples.

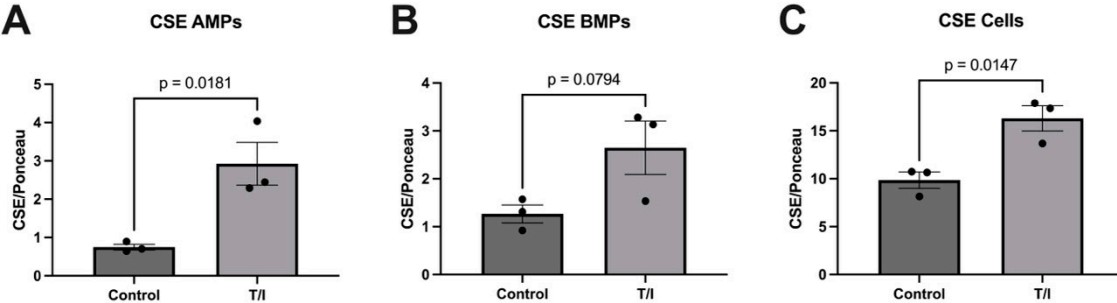

**Figure 1.** Analysis of CSE in apical microparticles (AMPs), basolateral microparticles (BMPs), and hCMEC/D3 human brain microvascular endothelial cells. Data are normalized to Ponceau staining. (**A**) CSE in control and cytokine (T/I)-treated AMPs derived from hCMEC/D3 human brain microvascular endothelial cells; (**B**) CSE in control and cytokine-treated BMPs derived from hCMEC/D3 human brain microvascular endothelial cells; (**C**) CSE in control and cytokine-treated hCMEC/D3 cell samples.

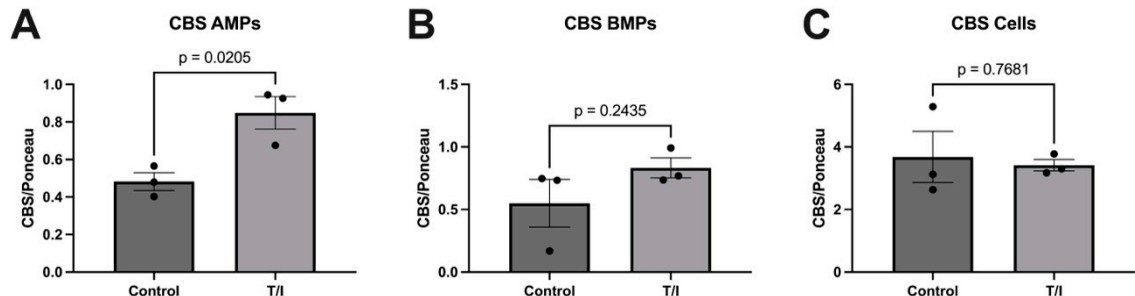

**Figure 2.** Analysis of CBS in apical microparticles (AMPs), basolateral microparticles (BMPs), and hCMEC/D3 human brain microvascular endothelial cells. Data are normalized to Ponceau staining. (**A**) CBS in control and cytokine (T/I)-treated AMPs derived from hCMEC/D3 human brain microvascular endothelial cells; (**B**) CBS in control and cytokine-treated BMPs derived from hCMEC/D3 human brain microvascular endothelial cells; (**C**) CBS in control and cytokine-treated hCMEC/D3 cell samples.

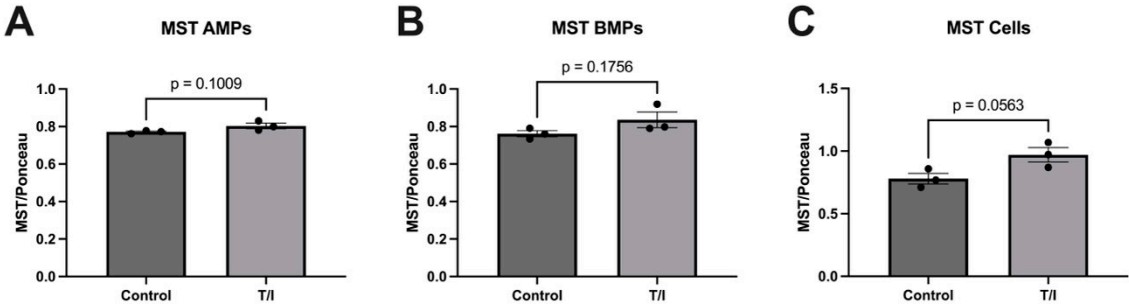

**Figure 3.** Analysis for MST in apical microparticles, basolateral microparticles, and human brain endothelial cells. Data are normalized from images following Ponceau staining and images. (**A**) MST in control samples and cytokine-treated apical microparticle samples derived from endothelial cells; (**B**) MST in control samples and cytokine-treated basolateral microparticles derived from endothelial cells; (**C**) MST in control and cytokine-treated D3 cell samples.

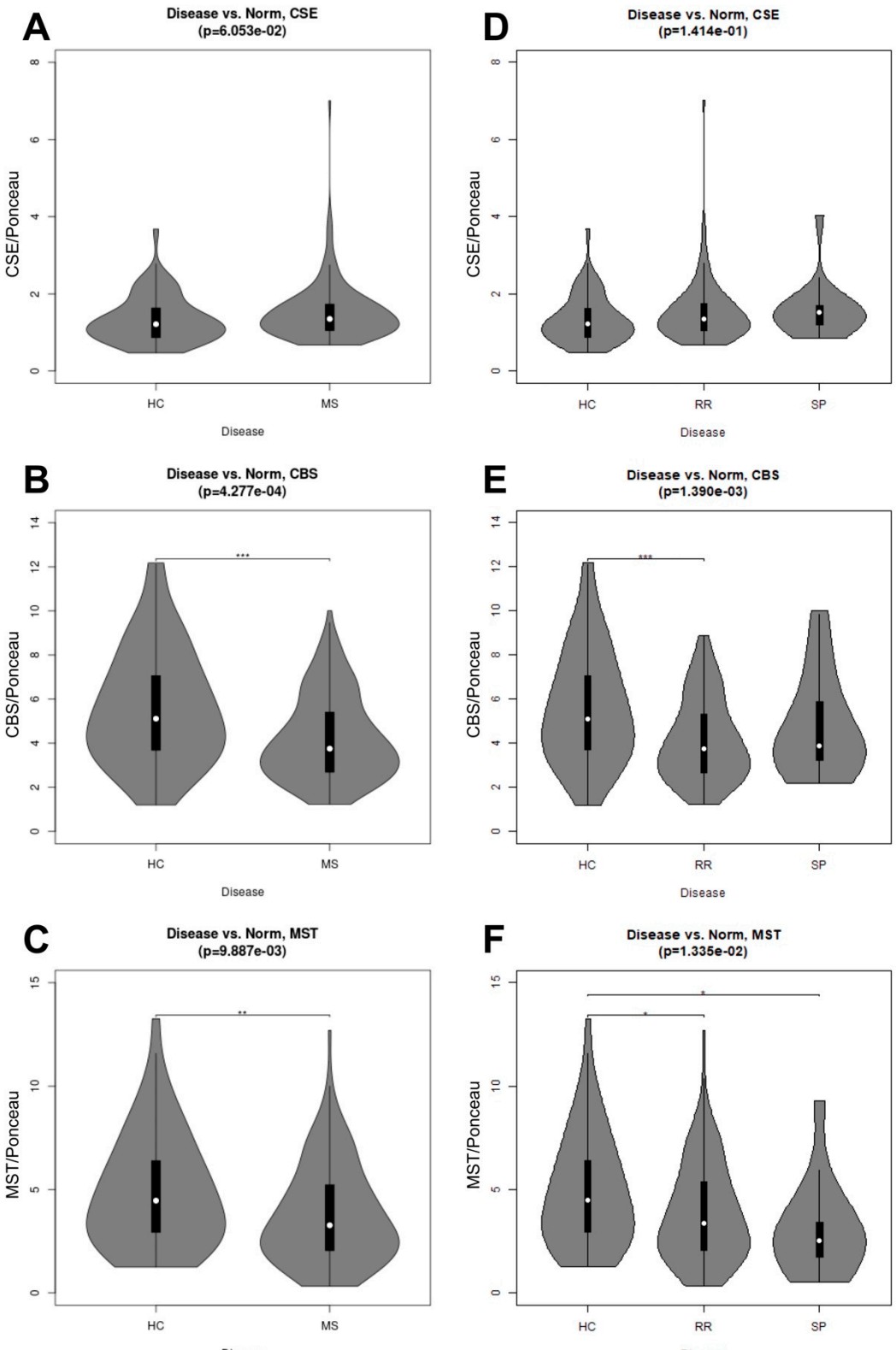

**Figure 4.** Serum blotting for CSE, CBS and MST in healthy controls (HC), in MS aggregate and MS subgroups (RRMS, SPMS). Violin Plots from Mann–Whitney U and Kruskal–Wallis Tests. Plots display the enzymatic changes of CSE (TOP ROW), CBS (MIDDLE ROW), and MST (BOTTOM ROW) in HC and condensed datasets (MS) (LEFT COLUMN), and HC and uncondensed datasets (RRMS vs. SPMS) (RIGHT COLUMN). Serum samples include 50 HC, 152 MS, 137 RRMS, and 15 SPMS. The white dot represents the median, while the box constitutes the interquartile range. (**A**) CSE in

HC vs. MS; (**B**) CBS in HC vs. MS; (**C**) MST in HC vs. MS; (**D**) CSE in HC vs. RRMS and SPMS; (**E**) CBS in HC vs. RRMS and SPMS; (**F**) MST in HC vs. RRMS and SPMS.

We also performed classification and regression tree (CART) analysis on the input data from the serum samples to produce a decision tree using the *rpart* software package for the R statistical language [23]. This decision tree predicts the disease state from the remaining variables (Figure 5). We used a complexity parameter of 0.01 and a minimum split of 20 with 10-fold cross-validation to produce the decision tree. The resulting visualization depicts the decision criteria, the number of records classified into each branch, and the accuracy of each branch.

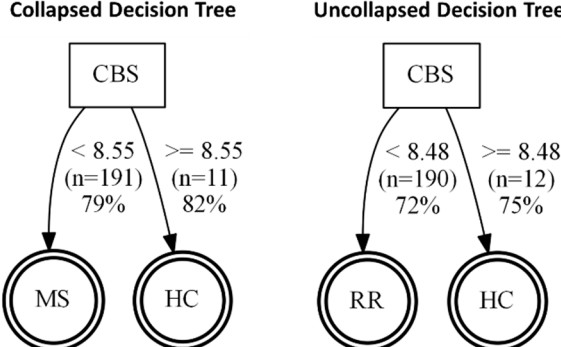

**Figure 5.** Diagram of decision trees for diagnostic evaluation of MS (left) and RRMS (right) based on CBS. Left—collapsed evaluation suggests that using CBS as a biomarker, MS can be discriminated from controls with 79 and 82% accuracy. Right—uncollapsed decision tree shows that CBS can also be used to discriminate RRMS from healthy controls with 72 and 75% accuracy, respectively.

## 3. Results

### 3.1. CSE Increased in Human Brain Endothelial Cells, AMPs, and BMPs in Response to TNF-α/IFN-γ Treatment

We observed that exposure of human brain endothelial cells to TNF-α/IFN-γ for 72 h significantly increased the overall cellular expression of the sulfide-producing enzyme CSE ($p = 0.0147$) by approximately 50%, and increased the release of CSE[+] AMPs by nearly threefold ($p = 0.0181$), while the release of CSE[+] BMPs was approximately doubled, although this did not reach statistical significance ($p = 0.0794$).

### 3.2. CBS Increased in Human Brain Endothelial AMPs in Response to TNF-α/IFN-γ Treatment

We observed that exposure of human brain endothelial cells to TNF-α/IFN-γ for 72 h significantly increased the release of CBS[+] AMPs ($p = 0.0205$). The release of CBS[+] BMPs increased as well but was not statistically significant ($p = 0.2345$). Expression of CBS in human brain endothelial cells unchanged by cytokine treatment.

### 3.3. MST was Not Significantly Changed in Human Brain Endothelial Cells in Response to TNF-α/IFN-γ Treatment

We observed that exposure of human brain endothelial cells to TNF-α/IFN-γ for 72 h produced a modest, but nonsignificant, increase in the cellular expression of MST in cells, but had minimal impact on MST[+] released into apical and basolateral microparticles (Figure 3).

It is worth noting that treatment with TNF-α or IFN-γ individually did not produce the same changes in CBS, CSE, and MST as those we observed when the cytokines were used in combination (data not shown), suggesting that these effects reflected combined TNF-α/IFN-γ stimulation.

### 3.4. CBS and MST were Decreased in MS Serum Samples in Comparison with HC

By comparison, we observed an increase in serum CSE between the healthy controls and combined MS groups (RRMS + SPMS). Similarly, we found increases in serum CSE between the healthy controls and RRMS and SPMS subgroups individually, but these did not reach statistical significance. On the other hand, CBS was significantly decreased in the MS group (Figure 4B) (*** $p$ = 0.0004). Statistical analysis of the MS subgroups only showed significance in the RRMS group (Figure 4E) (*** $p$ = 0.001). Similarly, MST decreased significantly in the MS serum samples when compared to the healthy controls (Figure 4C) (** $p$ = 0.009). Furthermore, analysis of MST in the MS subgroups also showed statistical significance between HC and both RRMS and SPMS subgroups (Figure 4F) (* $p$ = 0.01).

## 4. Discussion

Among multiple neurovascular and cardiovascular diseases, MPs have been investigated for their potential as biomarkers for disease states. MPs also influence disease progression through their ability to induce signaling in recipient or target cells. In previous research, endothelial MPs were shown to be produced more abundantly in response to inflammatory cytokine treatment [6]. We have previously observed that these human brain endothelial cells are polarized in nature and release AMPs and BMPs which are formed by apparently different cellular processes, and which carry different types of "cargo" derived from the parent cell lines [6]. In that study, exposure to inflammatory cytokines did not change AMP diameters; however, the production of BMPs was increased whereas the BMPs formed were smaller, which increased their total surface area to favor BMP binding to target cells. In the same study it was found that BMPs from cytokine-treated endothelial cells impaired contractility of brain vascular smooth muscle cells. Such effects could detrimentally influence neurovascular perfusion. Furthermore, AMPs from cytokine-treated cells disrupted brain endothelial barrier function. Thus, the release of microparticles into both the sub-endothelial or "perivascular" space and into the circulation under inflammatory conditions induce pathological changes (Figure 6). As part of these responses, changes in expression of CSE, CBS, and MST in the AMPs, BMPs, and human endothelial cells may contribute to the pathophysiology of MS and potentially other neurovascular conditions. While intriguing, we cannot claim that this model system exactly recapitulates the in vivo blood–brain barrier, but it may be a valuable first step towards understanding how these particles are formed and distributed in the vasculature.

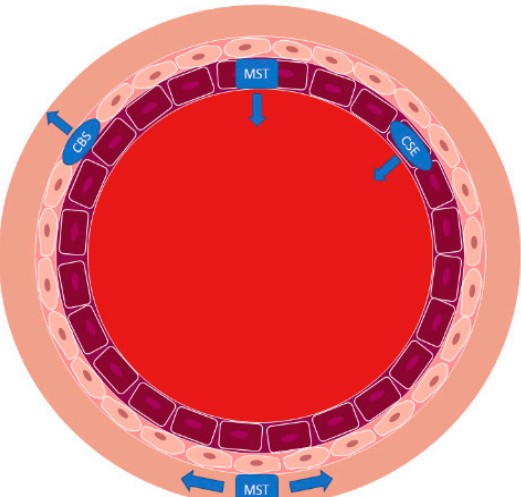

**Figure 6.** Diagram of blood vessel schematic of CSE, CBS, and MST within endothelial cells and smooth muscle. CSE is localized to the endothelial cells, CBS is primarily found in the smooth vascular muscle, and MST is seen in the surrounding vascular space and within the endothelial cells.

Based on prior investigations showing CSE, CBS, and MST in the vascular space, we explored whether inflammatory mediators increase expression of sulfide-generating enzymes in brain endothelial cells, as well as in their apically and basolaterally released MPs. Our blotting analysis results showed significant elevations in the expression of CSE in response to inflammatory cytokine stress (combined TNF/IFN—'T/I') compared with non-treated control samples. This increase in CSE expression in inflamed endothelial cells could help explain the elevation in circulating sulfides seen in neurodegenerative conditions, such as Alzheimer's disease, which has also been ascribed to CSE activity [9]. Importantly, the elevated sulfides in AD have been linked to the associated vascular stress and mediate half of the relationship between the microvascular abnormalities and cognitive impairment in this condition. These results suggest that in response to inflammatory stress, CSE is expressed by endothelial cells and is released into microparticles which may help drive elevated sulfides in AD and perhaps other neurodegenerative conditions. However, this is not reflected in MS, where the CSE levels were not elevated in the serum, thus revealing heterogeneity in sulfide metabolism between neurodegenerative diseases.

Our results also show that CBS may also be released from cells into both apical and basolateral MPs in response to cytokine treatment. This does not result in a loss of CBS from the endothelial cells, but rather the cells maintained a steady state of this enzyme. In contrast, the MST levels remained largely unchanged, with just minor increases in all samples. The fact that both CSE and CBS are released into MPs on both sides of the endothelial cells suggests that the expression of both enzymes in the blood may reflect release of CSE and CBS towards the brain, into the perivascular space. Together, our in vitro data highlight the differential regulation of expression and packaging into MPs of the three sulfide-generating enzymes in brain endothelial cells. We suggest further analysis should be performed to better understand their contributions to neurovascular/neurodegenerative diseases that have inflammatory components.

Once we determined that cytokines can regulate the release of sulfide-generating enzymes, in particular CBS and CSE, into the "vascular" space, i.e., AMPs, we next tested whether any of the enzymes may be suitable biomarkers for MS. Next, we sought to determine whether our findings from our brain endothelial cell study reflect what is seen in the serum of patients with MS. However, we saw only a modest increase in CSE expression ($p = 0.0605$) in MS (both RRMS and SPMS considered together as "collapsed"). When RRMS and SPMS were separated, this trend was not significant. We next considered changes in CBS expression and found remarkable reductions in CBS in the "collapsed" MS (RRMS+SPMS), which was still found to be significant when SPMS and RRMS were considered individually ("uncollapsed"). Using a decision tree analysis, this surprising reduction in serum CBS levels (Figure 5) was found to predict MS disease activity, and specifically RRMS disease activity, with 82% and 75% predictive accuracy, respectively. While MST was also significantly reduced in "collapsed" and "uncollapsed" MS, we found that considering MST as a diagnostic "tool" did not improve the predictive power for anticipating MS over that of CBS alone. Taken together, our results support that modest mobilization of CSE, and "shedding" into serum, as well as a very significant reduction in circulating CBS (and MST), may indicate a signature pattern of sulfide dysregulation in MS. While there may be a role suggested for elevated CSE in AD, here we found that reductions in CBS, and to a lesser extent MST, better characterize, and may influence, MS disease activity.

We hypothesize that vascular dysfunction in MS appears to involve diminished levels of CBS and MST in the brain [24] which may increase oxidative stress in the neurovasculature. In MS, increased brain oxidative stresses, particularly within the vascular and parenchymal spaces, could intensify disease progression through a reduction in protective sulfides generated in different brain "compartments" (neural/glial/vascular) which fail to limit oxidant clearance. Inflammatory stresses could therefore dysregulate brain sulfides in MS patients, who typically have a high burden of reactive oxidative species

that may exacerbate cerebrovascular stress. Reductions in these enzymes could also diminish circulating sulfides to decrease cerebral blood flow as a mechanism of MS disease activity.

## 5. Conclusions

This study evaluated changes in protein levels of CSE, CBS, and MST in a polarized, immortalized human brain endothelial cell model and the apical and basolateral MPs generated by the cells [25] with the goal of determining how sulfide-producing enzymes may govern the initiation and progression of RRMS and SPMS. MS is a neurodegenerative and neurovascular disease whose exacerbations are commonly associated with "cytokine storm" and inflammation in the central nervous system. The progression of MS disrupts brain vascular functions leading to inflammatory responses in the central nervous system, which results in vascular injury in the brain that can intensify neurodegeneration. There are limited treatments available to slow down or arrest the neurodegenerative processes in patients with RRMS and SPMS, with better efficacy when treatment is started in the early stages of the disease. Therefore, identification of early biomarkers is critical for therapeutic intervention. Prior evidence of proinflammatory cytokines initiating the release of endothelial AMPs and BMPs [6] prompted us to investigate the presence of sulfide-producing enzymes in brain endothelial cells and their MPs. The discrepancies we see between the protein levels of CSE, CBS, and MST in serum seen in vivo versus those released by brain endothelial cells in vitro suggest that endothelial cells may contribute to the CSE elevations, but perhaps not to the reductions in CBS and MST. Therefore, endothelial responses may not be the only source influencing the results seen in MS patient serum. Alternatively, the differences may be due to the acute nature of the cell culture model versus the chronic timeline of the disease. Additionally, different cellular sources such as nerves, glia, and smooth muscle may represent alternative sources of these enzymes, which would affect serum levels.

Identification of CSE, CBS, and MST in serum, human brain endothelial cells, and their AMPs and BMPs now permits investigation of sulfide-producing enzymes in MS and other forms of neurodegeneration using inexpensive and simple serum testing. The dramatic reductions in CBS and MST suggest that, rather than endothelial cells, neuronal and mitochondrial sulfide hypo-expression may be important in MS pathophysiology. Conversely, the observation that CSE expression is also increased in serum and in endothelial cells could influence MS but to a lesser extent. By developing standardized tests for MS using these biomarkers, MS might be detected prior to advanced neurodegeneration, prompting treatment to arrest MS progression earlier.

While this study found remarkable changes in serum sulfide-producing enzymes, it is not yet clear whether the products of these enzymes, namely circulating sulfides, are in fact elevated in MS, as they have been reported to be increased in AD [9]. Future studies comparing the serum levels of sulfides with their synthetic machinery may provide a clearer picture of these relationships. However, we have now demonstrated successfully for the first time that reductions in serum CBS (and MST) appear to represent useful markers of MS disease activity.

**Author Contributions:** Conceptualization, P.V., K.Y.S., E.D., C.G.K., and J.S.A.; methodology, J.W.Y., M.R., R.Z., P.V., and J.S.A.; validation, P.V., M.R.,and R.Z.; formal analysis, P.V., J.W.Y., U.C., M.T., N.D, and P.K.; investigation, P.V., M.R., R.Z., and J.S.A.; resources, J.W.Y., M.R., R.Z., and J.S.A.; writing—original draft preparation, P.V., N.D., and J.S.A.; writing—review and editing, P.V., N.D., J.W.Y., K.Y.S., E.D., C.G.K., U.C., M.T., P.K., M.R., R.Z., and J.S.A.; visualization, P.V., N.D., U.C., N.D. and P.K.; supervision, J.S.A.; project administration, J.S.A. All authors have read and agreed to the published version of the manuscript.

**Funding:** This work was supported in part by the Center for Redox Biology and Cardiovascular Disease Center of Biomedical Research Excellence supported by the National Institute of General

Medical Sciences IDeA program of the National Institutes of Health under award P20GM121307 and by the IDeA Center of Biomedical Research Excellence grant P20GM103424.

**Institutional Review Board Statement:** The study was conducted in accordance with the Declaration of Helsinki, and approved by the Institutional Review Board (IRB ('MS-Omics project RQ00328', HSIRB 435893-2, approved 9/6/13).

**Informed Consent Statement:** Informed consent was obtained from all subjects involved in the study.

**Data Availability Statement:** The data presented in this study are available on request from the corresponding author.

**Conflicts of Interest:** The authors declare no conflict of interest

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
