# Peer review of "Dysregulated Sulfide Metabolism in Multiple Sclerosis: Serum and Vascular Endothelial Inflammatory Responses"

_pathophysiology, doi:10.3390/pathophysiology29030044_

Round 1

Reviewer 1 Report

The manuscript by P. Veerareddy et al entitled “Dysregulated Sulfide Metabolism in Multiple Sclerosis: Serum and Vascular Endothelial Inflammatory Responses” is presented for review.

The purpose of this study was to evaluate changes in the sulfide generating enzymes (CSE, CBS, and MST) in brain endothelial cells in culture and in apical and basolateral microparticles (AMPs, BMPs) derived from these cells when exposed to inflammatory cytokines (e.g. TNF-a and IFN-g; as a model of MS inflammation) and compare these responses with the CSE, CBS, and MST enzyme levels in serum of MS patients. The overall goal of the study was to determine how sulfide-producing enzymes may govern the initiation and progression of RRMS and SPMS and serve as markers of MS disease activity.

The obtained results indicate increase in CSE and CBS in both AMP and BMP produced by TNF-a/IFN-g-stimulated hBMEC/D3 cells and a modest mobilization of CSE, and ‘shedding’ into serum as well as a very significant reduction in circulating CBS (and MST) that may indicate a signature pattern of sulfide dysregulation in MS.

It is this reviewer’s opinion that the study is focused, novel, addresses major clinical problem, and may have a significant impact to the field. 

The manuscript is very well written and was easy to follow. The results for the most part are self-explanatory, clearly presented, and well discussed. Potential caveats/limitations are acknowledged and addressed. 

As such, this reviewer has NO MAJOR concerns.

Minor concerns:

1) The abstract could be more structured to improve readability. 

2) Methods section (Line 143): It is indicated that serum and plasma samples were collected, while only serum samples were used in this study. Please clarify.

3) Methods section (Line 168): Please indicate membrane pore size of the 6-well cell culture inserts used in this study. 

4) Does apical only stimulation of hBMEC/D3 with TNF-a/IFN-g induces production of BMPs?

5) What could be explanation for the observed increase in CBS (in both AMP and BMP) in TNF-a/IFN-g-stimulated hCMEC/D3 cells in vitro (Figures 2A/B) vs a decrease in CBS levels in serum of MS patients (Figure 4B)?

Author Response

The manuscript by P. Veerareddy et al entitled “Dysregulated Sulfide Metabolism in Multiple Sclerosis: Serum and Vascular Endothelial Inflammatory Responses” is presented for review.

The purpose of this study was to evaluate changes in the sulfide generating enzymes (CSE, CBS, and MST) in brain endothelial cells in culture and in apical and basolateral microparticles (AMPs, BMPs) derived from these cells when exposed to inflammatory cytokines (e.g. TNF-a and IFN-g; as a model of MS inflammation) and compare these responses with the CSE, CBS, and MST enzyme levels in serum of MS patients. The overall goal of the study was to determine how sulfide-producing enzymes may govern the initiation and progression of RRMS and SPMS and serve as markers of MS disease activity.

The obtained results indicate increase in CSE and CBS in both AMP and BMP produced by TNF-a/IFN-g-stimulated hBMEC/D3 cells and a modest mobilization of CSE, and ‘shedding’ into serum as well as a very significant reduction in circulating CBS (and MST) that may indicate a signature pattern of sulfide dysregulation in MS.

It is this reviewer’s opinion that the study is focused, novel, addresses major clinical problem, and may have a significant impact to the field. 

The manuscript is very well written and was easy to follow. The results for the most part are self-explanatory, clearly presented, and well discussed. Potential caveats/limitations are acknowledged and addressed. 

As such, this reviewer has NO MAJOR concerns.

Minor concerns:

  • The abstract could be more structured to improve readability. 

Response. We agree and have now restructured the abstract

  • Methods section (Line 143): It is indicated that serum and plasma samples were collected, while only serum samples were used in this study. Please clarify.

Response. We have changed all mentions of plasma to serum.

  • Methods section (Line 168): Please indicate membrane pore size of the 6-well cell culture inserts used in this study. 

Response. We have now indicated that the size of the pores is 3 µm

  • Does apical only stimulation of hBMEC/D3 with TNF-a/IFN-g induces production of BMPs?

Response. We appreciate this comment. We do not know if apical stimulation induces production of BMPs as we added treatment to both the top and bottom compartments. Future studies may consider this.

  • What could be explanation for the observed increase in CBS (in both AMP and BMP) in TNF-a/IFN-g-stimulated hCMEC/D3 cells in vitro (Figures 2A/B) vs a decrease in CBS levels in serum of MS patients (Figure 4B)?

Response. This is a good point. In lines 378-383, we address this point and have added some additional discussion beyond this.

Reviewer 2 Report

I reviewed the paper by Veerareddy et al. with great interest. The authors sought to investigate potential changes in enzymes of the endothelial hydrogen sulfide (H2S) pathway in response to inflammatory cytokines, and to determine if changes in 3 key enzymes are altered in MS patients.  The questions are relevant, and the methods sound. I have minor comments to improve the manuscript.

1. The differential responses observed in culture versus serum/plasma require additional explanation in the discussion, including the likelihood of non-brain vascular sources.

2. Both serum and plasma are listed. When was one versus the other used in experiments, and would the MPs/enzymes be different between preparations?

3. Why did the authors use blotting instead of ELISA; particualrly for plamsa measurents?

4. Can the authors confirm that the cells, which are immortalized, maintain polarity?

5. Can the authors clarify why the CBS experiements using cultured cells do not show an increase in the cell prep?

6. Additional discussion of how cerebral blood flow might be altered secondary to enzyme changes would be appreciated.

Author Response

I reviewed the paper by Veerareddy et al. with great interest. The authors sought to investigate potential changes in enzymes of the endothelial hydrogen sulfide (H2S) pathway in response to inflammatory cytokines, and to determine if changes in 3 key enzymes are altered in MS patients.  The questions are relevant, and the methods sound. I have minor comments to improve the manuscript.

  1. The differential responses observed in culture versus serum/plasma require additional explanation in the discussion, including the likelihood of non-brain vascular sources.

Response. This is a good point, also mentioned by reviewer 1. We have now addressed this in lines 378-385, where other enzyme sources are suggested.

  1. Both serum and plasma are listed. When was one versus the other used in experiments, and would the MPs/enzymes be different between preparations?

Response. We regret this error. We have eliminated all mentions of plasma.

  1. Why did the authors use blotting instead of ELISA; particualrly for plamsa measurents?

Response. This is a good point. There are no commercial ELISA for these enzymes.

  1. Can the authors confirm that the cells, which are immortalized, maintain polarity?

Response. This is an excellent point. Although immortalized, the D3 cell line has been studied extensively as a model of the blood brain barrier, which is polarized (Weksler, B., Romero, I.A. & Couraud, PO. The hCMEC/D3 cell line as a model of the human blood brain barrier. Fluids Barriers CNS 10, 16 (2013). https://doi.org/10.1186/2045-8118-10-16). We now mention this in line 189 and added this citation to address this concern.

  1. Can the authors clarify why the CBS experiements using cultured cells do not show an increase in the cell prep?

Response. This is a good point. As we mentioned to Reviewer 1, this appears to reflect different cellular sources of these enzymes, which appear in serum samples. This is described in lines 378-385.

  1. Additional discussion of how cerebral blood flow might be altered secondary to enzyme changes would be appreciated.

Response. This is a good point. We have addressed this in lines 365-366, where we say “Reductions in these enzymes could diminish circulating sulfides to decrease cerebral blood flow as a mechanism of MS disease activity.”

Reviewer 3 Report

I was impressed by the article ‘Dysregulated Sulfide Metabolism in Multiple Sclerosis: Serum and Vascular Endothelial Inflammatory Responses' by Veerareddy et al. It is a very interesting, well written study. The authors evaluated the levels of CSE, CBS, and MST in serum samples from MS patients and healthy controls. At the end of the study, they reported that they found a decrease in serum CBS and MST levels and an increase in CSE levels in MS patients.

Actually, the work is well planned and written. However, if the authors have adding some parameters can increase the value of the article. For example, if possible, if correlation analysis of CSE, CBS and MST values ​​can be done, perhaps more important data can be obtained.

It is sufficient in the summary section, the aims, methods and results of the study are given adequately and statistical data are added.

The introduction is clear and understandable and the material and method are well written. The introduction is written clearly and clearly and the purpose is stated.

In the material and method section, normal brain cells were studied. Doing the same study with brain cells with MS would have been better to evaluate the results. Because sometimes the results can be different in diseased cells. Authors can take this into account when planning new projects. In addition, immunohistochemical studies can also provide important data. However, this is a study in which important results were obtained.

In results section, presentation of findings and tables are given appropriately and are adequate.

Discussion, conclusion and references are enough. The language is also very fluent and easy to understand. I recommend minor revision.

Author Response

I was impressed by the article ‘Dysregulated Sulfide Metabolism in Multiple Sclerosis: Serum and Vascular Endothelial Inflammatory Responses' by Veerareddy et al. It is a very interesting, well written study. The authors evaluated the levels of CSE, CBS, and MST in serum samples from MS patients and healthy controls. At the end of the study, they reported that they found a decrease in serum CBS and MST levels and an increase in CSE levels in MS patients.

Actually, the work is well planned and written. However, if the authors have adding some parameters can increase the value of the article. For example, if possible, if correlation analysis of CSE, CBS and MST values ​​can be done, perhaps more important data can be obtained. 

Response. This is a very good idea. We have future studies planned to consider this but feel that it is outside the scope of the current study.  

It is sufficient in the summary section, the aims, methods and results of the study are given adequately and statistical data are added.

The introduction is clear and understandable and the material and method are well written. The introduction is written clearly and clearly and the purpose is stated.

In the material and method section, normal brain cells were studied. Doing the same study with brain cells with MS would have been better to evaluate the results. Because sometimes the results can be different in diseased cells. Authors can take this into account when planning new projects. In addition, immunohistochemical studies can also provide important data. However, this is a study in which important results were obtained.

Response. We appreciate these helpful comments and do plan to evaluate this in tissue specimens in the future.

In results section, presentation of findings and tables are given appropriately and are adequate.

Discussion, conclusion and references are enough. The language is also very fluent and easy to understand. I recommend minor revision.

Response. We appreciate the enthusiastic appraisal of our study.